# Astragalus in Acute Pancreatitis: Insights from Network Pharmacology, Molecular Docking, and Meta-Analysis Validation

**DOI:** 10.3390/cimb47050379

**Published:** 2025-05-21

**Authors:** Xingxin Cao, Suqin Duan, Aiyi Li, Zhanlong He

**Affiliations:** 1Institute of Medical Biology, Chinese Academy of Medical Sciences & Peking Union Medical College, Kunming 650118, China; s2024018001@student.pumc.edu.cn (X.C.); duansuqin129@163.com (S.D.); 2College of Life Sciences, Yunnan University, Kunming 650500, China; liaiyi2000@163.com

**Keywords:** *Astragalus*, acute pancreatitis, network pharmacology, molecular docking, meta-analysis

## Abstract

(1) Backgroud Astragalus, a traditional Chinese medicine, demonstrates therapeutic effectiveness in acute pancreatitis (AP). Nevertheless, its precise pharmacological mechanism remains unclear, and clinical guidelines have not been established. This study aims to systematically elucidate the active compounds and molecular mechanisms underlying Astragalus’ therapeutic effects in AP, and provide clinical evidence supporting its efficacy. (2) Methods: TCMSP and Swiss Target Prediction identified drug targets; GeneCards, DrugBank, and OMIM provided disease targets. Venny determined the therapeutic targets, while STRING constructed a protein–protein interaction network. Cytoscape 3.10.3 validated core targets. DAVID was used to conduct GO and KEGG pathway analyses, visualized via Bioinformatic platform. Cytoscape 3.10.3 was used to build a “drug–ingredients–targets–pathways–disease” network. AutoDock Vina 1.1.2 and AutoDockTools 1.5.7 was used to performed molecular docking, with PyMOL 3.0 visualizing the results. PubMed, Embase, Cochrane, Web of Science, CNKI, Wanfang, VIP, and CBMdisc were searched. The literature was screened, extracted, and evaluated, followed by a meta-analysis, using RevMan 5.4.1 and Stata 18. (3) Results: We identified 539 targets for the active ingredients of astragalus. Among 1974 disease-related targets, 232 were found to be therapeutic targets. The GO analysis yielded 589 entries, while the KEGG pathway enrichment analysis identified 147 relevant pathways. The top five active ingredients were quercetin, kaempferol, isorhamnetin, formononetin, and calycosin. Molecular docking analysis revealed potential synergistic effects between these components and core targets. The meta-analysis, comprising six randomized controlled trials, demonstrated a significantly higher total effective rate of clinical efficacy in the astragalus group compared to the control group. (4) Conclusions: Astragalus treats AP through the synergistic action of its components, targets, and pathways. Key active compounds, such as quercetin, kaempferol, isorhamnetin, formononetin, and calycosin, engage with pivotal targets, including TP53, AKT1, TNF, IL6, EGFR, CASP3, MYC, and HIF1A, within primary pathways, such as pathways in cancer, PI3K-Akt signaling pathway, and lipid metabolism, and atherosclerosis. Astragalus effectively treats AP and alleviates clinical symptoms by reducing the time for gas or defecation passage, the disappearance time of abdominal pain or distension, and the recovery time of bowel sounds.

## 1. Introduction

Acute pancreatitis (AP) is a prevalent gastrointestinal disorder frequently resulting in hospitalization [1]. Characterized by pancreatic autodigestion and potential damage to distant organs, it can progress to metabolic diseases such as diabetes, with an incidence of 110 to 140 per 100,000 individuals and a mortality rate of 1–5% [2,3]. AP is an acute pancreatic inflammation, primarily triggered by metabolic or infectious factors that activate enzymes in pancreatic acinar cells, leading to self-digestion and potentially necrotizing inflammation both locally and systemically [4]. The urgent need for effective therapeutic strategies is underscored by the current lack of treatments and the significant threat AP poses to human health [5].

In recent years, Chinese traditional medicines (TCMs) have demonstrated therapeutic efficacy in AP, with various TCM herb decoctions being utilized [6,7]. Astragalus has notably emerged as a prominent single herb in these treatments. According to the specifications outlined in the Chinese Pharmacopoeia, medical astragalus (Huangqi in Chinese) refers to the dehydrated roots of the following two plants, which are extensively used in traditional Chinese medicine: *Astragalus membranaceus* (Fisch.) Bge. Var. *mongohlicus*, (Bge.) Hsiao, and *Astragalus membranaceus* (Fisch.) Bge. (Fam. Leguminosae) [8]. Clinical trials have demonstrated that astragalus injection, an intravenous drug derived from astragalus, exhibits significant coadjuvant therapeutic efficacy in AP [9].

Nevertheless, the utilization of astragalus for AP is constrained by the lack of clarity regarding its pharmacological mechanism and the absence of established clinical guidelines, systematic reviews, or meta-analyses. In this study, we aim to elucidate the potential pharmacological mechanisms through network pharmacology and conduct a meta-analysis to establish a pharmacological and evidence-based rationale for the application of astragalus in AP.

## 2. Materials and Methods

### 2.1. Bioactive Ingredients and Action Targets of Astragalus

The herb “astragalus” was analyzed using the Traditional Chinese Medicine Systems Pharmacology (TCMSP, https://www.tcmsp-e.com, accessed on 12 May 2025) and the Swiss Target Prediction platform (http://swisstargetprediction.ch, accessed on 12 May 2025). Bioactive ingredients were identified by applying criteria including a drug-likeness (DL) value of at least 0.18 and oral bioavailability (OB) exceeding 30%. The protein names corresponding to the bioactive ingredients were converted to gene names utilizing the Uniprot database (https://www.uniprot.org, accessed on 12 May 2025).

### 2.2. Identification of AP Disease Targets

The term “acute pancreatitis” was queried across the GeneCards (https://www.genecards.org, accessed on 12 May 2025), DrugBank (https://go.drugbank.com, accessed on 12 May 2025), and Online Mendelian Inheritance in the Man (OMIM, https://www.omim.org, accessed on 12 May 2025) databases. The search results from GeneCards were filtered based on a relevance score greater than 10.

### 2.3. Acquisition of Therapeutic Targets

The therapeutic targets of astragalus for treating AP were determined by identifying the intersection between drug targets and disease targets using the Venny platform (https://bioinfogp.cnb.csic.es/tools/venny/index.html, accessed on 12 May 2025).

### 2.4. Construction of Protein–Protein Interaction Network and Identification of Core Targets

The protein–protein interaction (PPI) network was constructed using the therapeutic targets identified in the previous step, based on data from the STRING database (https://cn.string-db.org, accessed on 12 May 2025). Core targets were subsequently identified by assessing their degree, betweenness, and closeness values using the CentiScape 2.2 tool within Cytoscape 3.10.3. The core targets were selected based on their median values across the three parameters, with the screening results ranked according to their degree values.

### 2.5. GO Enrichment Analysis and KEGG Pathway Analysis

The Gene Ontology (GO) enrichment analysis and Kyoto Encyclopedia of Genes and Genomes (KEGG) pathway analysis were performed by uploading therapeutic targets to the DAVID database (https://davidbioinformatics.nih.gov, accessed on 12 May 2025). Only the data with a *p*-value <0.01 were included. Subsequently, the GO and KEGG figures were generated using Bioinformatics platform (http://www.bioinformatics.com.cn, accessed on 12 May 2025). The “drug–ingredients–targets–pathways–disease” topological network was constructed using previous data in Cytoscape 3.10.3.3 (Institute for Systems Biology, Seattle, WA, USA). 

### 2.6. Molecular Docking

The top five active ingredients of astragalus were pinpointed using the “drug–ingredients–targets–pathways–disease” network, followed by molecular docking with core target proteins. The SDF files of these active ingredients were sourced from the TCSMP database and optimized for minimum free energy conformation, using the “Calculations–MM2–Minimize Energy” function, and then saved as MOL2 files in Chem3D 22.0.0. (CambridgeSoft Corporation, Cambridge, MA, USA). Research Collaboratory for Structural BioinformaticsRCSB database (RCSB, https://www.rcsb.org, accessed on 12 May 2025). The target protein structures were then prepared by removing waters, organic molecules, and ligands using PyMOL 3.0 (Schrödinger, South San Francisco, CA, USA).

We performed molecular docking using AutoDock Vina 1.1.2 (Scripps Research, La Jolla, CA, USA) and AutoDockTools 1.5.7 (Scripps Research, La Jolla, CA, USA), visualizing the results with PyMOL. The target protein was designated as a macromolecule and saved in the PDBQT format by selecting “Grid–Macromolecular–Choose”. The ligand was also input and saved in the PDBQT format. The grid box was constructed by setting the spacing to 1 Å and the *X*-, *Y*-, and *Z*-dimensions to precisely encompass the macromolecule, with the parameters saved in the GPF format. Utilizing the macromolecule, ligand, and grid box parameters, the molecular docking simulation was performed using AutoDock Vina 1.1.2.

### 2.7. Meta-Analysis

#### 2.7.1. Literature Retrieval

A comprehensive search was conducted in multiple databases, including PubMed, Embase, Cochrane Library, Web of Science, Chinese National Knowledge Infrastructure (CNKI), Wanfang, VIP, and China Biology Medicine disc (CBMdisc), to identify relevant literature on the utilization of astragalus in AP treatment. The search covered articles published up to 11 June 2024, using the keywords “Astragalus” and “Huangqi” for the drug, and “acute pancreatitis” for the disease. This study was prospectively registered on the PROSPERO platform (CRD42024627711, https://www.crd.york.ac.uk/PROSPERO; accessed on 12 May 2025).

#### 2.7.2. Study Selection

The included studies met the following criteria: (1) The study population consisted of patients with AP who were accurately diagnosed according to established guidelines. (2) The study design was a randomized controlled trial (RCT), with participants randomly assigned to either a test group receiving astragalus injection, in addition to general treatment, or a control group receiving general treatment. (3) The outcome measures included the ratio of effective treatment, time for passage of gas or defecation, disappearance time of abdominal pain or distension, and recovery time of bowel sounds.

The following studies were excluded: (1) nonclinical controlled trials, such as retrospective analyses, reviews, and animal studies; (2) duplicate publications using the same patient data; (3) incomplete research data that could not be traced back to the source; and (4) clearly flawed trial designs affecting data accuracy.

#### 2.7.3. Date Extraction

The extracted data from the included studies comprised the following: (1) first author’s name and the publication year; (2) characteristics and sizes of the test and control groups; (3) intervention details for both groups, including treatment duration, dosage, frequency, administration methods of drugs, and general treatment specifics; (4) outcome indices, including the effective treatment ratio, time to gas passage or defecation, disappearance time of abdominal pain or distension, and recovery time of bowel sounds.

#### 2.7.4. Methodological Quality Assessment

Following the risk of bias assessment tools recommended by the Cochrane Library, the methodological quality of the included RCTs was evaluated across seven domains, encompassing selection bias (random sequence generation and allocation concealment), performance bias, detection bias, attrition bias, reporting bias, and other biases, utilizing RevMan 5.4.1. Two independent researchers carried out the entire assessment process, with any discrepancies resolved through collaborative discussion.

#### 2.7.5. Statistical Analysis

The statistical analysis was conducted using RevMan 5.4.1. 1 (Epic Gecko Ltd., Bergen, Norway) and Stata 18 (StataCorp LLC, College Station, TX, USA). Heterogeneity was assessed using the Cochrane *Q* index and *I*^2^ index. A random-effects model was applied in the presence of significant heterogeneity (*I*^2^ ≥ 50%, *p* < 0.05), while a fixed-effects model was employed in its absence (*I*^2^ < 50%, *p* ≥ 0.05). The odds ratio (*OR*) was utilized as the combined effect measure for the categorical data, and the standardized mean difference (*SMD*) was used for the continuous data, both reported with their corresponding 95% confidence intervals (*CIs*).

## 3. Results

### 3.1. Retrieval Results of Active Ingredients of Astragalus

From the 87 ingredients of astragalus, 20 active ingredients were identified using the TCMSP database (Table 1) By converting the protein names of the targets associated with the 20 active ingredients to gene names through the UniProt database, 195 targets were obtained. Additionally, 424 targets were discovered by uploading the 2D structure SDF files of the active ingredients to the Swiss Target Prediction platform. After eliminating duplicate and invalid targets, a total of 539 unique targets were compiled.

### 3.2. Acquisition of Ap Targets

The term “Acute Pancreatitis” was queried in the GeneCards, OMIM, and DrugBank databases. A total of 1670 targets were identified by filtering for relevance scores greater than 10 in the GeneCards database. OMIM yielded 177 targets, while DrugBank provided 6 targets. Upon the integration of data from all three databases and the removal of duplicates, a total of 1794 unique targets were identified.

### 3.3. Validation of Therapeutic Targets

A total of 232 therapeutic targets were identified through the intersection of drug targets and disease targets using the Venny platform (Figure 1).

### 3.4. Construction of PPI Network

The interactions among the 232 therapeutic targets were analyzed using the STRING database to construct a PPI network (Figure 2). Each gene is represented by a distinct node, with interactions depicted by solid lines in red, purple, black, and green, indicating gene proximity, estrangement, co-occurrence, and co-expression, respectively. Core targets were identified by analyzing STRING data with the CentiScape 2.2 tool in Cytoscape 3.10.3, based on the median degree, betweenness, and closeness values, resulting in 50 nodes and 1098 edges (Figure 3) The significance of the core targets is highlighted by a color gradient from yellow to purple and an increased node size. Notably, key targets, such as TP53, AKT1, TNF, IL6, EGFR, CASP3, MYC, and HIF1A, play pivotal roles in the network (Table 2).

### 3.5. GO Enrichment Analysis and KEGG Pathway Analysis

The GO term analysis and KEGG pathway analysis were performed using the DAVID database and visualized via the Bioinformatics platform.

The GO enrichment analysis results were filtered using a *p*-value threshold of <0.01 and sorted by gene generation. The top ten biological processes (BPs), cellular components (CCs), and molecular functions (MFs) were compiled into a combined histogram (Figure 4). The analysis revealed that in terms of BPs, the key functions included the positive regulation of transcription by RNA polymerase II, negative regulation of apoptotic processes, and positive regulation of gene expression. The CCs are predominantly located in the cytoplasm, nucleoplasm, and cytosol, while the MFs are chiefly associated with protein binding, identical protein binding, and enzyme binding during astragalus treatment of AP.

Using a screening criterion of *p* < 0.01 and evaluating the gene counts, we identified 20 key pathways from 147 signaling pathways via DAVID for a bubble diagram (Figure 5). Noteworthy pathways, such as pathways in cancer, lipid metabolism, and atherosclerosis, as well as the PI3K-Akt signaling pathway, were found to be crucial for the overall treatment progress. The “drug–ingredients–targets–pathways–disease” network analysis highlighted highly enriched targets, including AKT1, TP53, and TNF, and pathways, such as pathways in cancer (hsa05200), lipid metabolism and atherosclerosis (hsa05417), and the PI3K-Akt signaling pathway (hsa04151) (Figure 6). Additionally, quercetin (MOL000098), kaempferol (MOL000422), isorhamnetin (MOL000354), formononetin (MOL000392), and calycosin (MOL000417), emerged as the top five active ingredients in this context.

### 3.6. Molecular Docking Results

Using AutoDock Vina 1.1.2, quercetin, kaempferol, isorhamnetin, formononetin, and calycosin were docked with receptor proteins based on 3D structures of core targets. The binding energies and modes for each complex were calculated. The results indicate that the binding energies for these active components with the core target proteins were below zero, suggesting potential interactions (Figure 7).

To improve the interpretation of the docking poses and scores, quercetin—the active compound in astragalus, affecting the most targets for treating AP—was compared with a known ligand for the analyzed proteins (Table 3). Visualization of the binding mode, using PyMOL 3.0, demonstrates that quercetin forms hydrogen bonds with each receptor protein (Figure 8).

### 3.7. Meta-Analysis

#### 3.7.1. Literature Retrieval

A total of 211 studies were retrieved from databases, with no additional studies found from other sources. Following the removal of 118 duplicate studies, 25 articles were screened based on their titles and abstracts to exclude animal experiments, clinical case reports, reviews, comments, and survey reports. Ultimately, 6 studies met the eligibility criteria for inclusion in the meta-analysis after excluding those involving non-astragalus therapeutic measures and those incompatible with the research indicators and objectives upon full-text evaluation [9,10,11,12,13,14] (Figure 9).

#### 3.7.2. Data Extraction and Methodological Assessment

The key characteristics of the six included studies are summarized in Table 4. Patients diagnosed with AP according to established guidelines were randomly assigned to either the astragalus or control group, with no significant differences in the baseline data. Astragalus injection, administered subcutaneously or intravenously, was the primary intervention for the astragalus group. Outcome indicators were measured systematically, with complete data records. Methodologically, the Jadad scores of the included studies ranged from 2 to 4 and were reassessed and visualized using RevMan 5.4.1 (Figure 10). It is noted that performance and detection biases were high due to the lack of double-blind trial designs in the included studies.

#### 3.7.3. Meta-Analysis Results of Astragalus for AP

Treatment effective of Astragalus

A total of 619 patients were included across six studies, with 312 in the astragalus group and 309 in the control group. A fixed-effects model was employed due to the absence of significant heterogeneity (*I*^2^ = 7%, *p* = 0.37). The meta-analysis indicates that astragalus is effective in treating AP [*OR* = 3.13, 95%*CI* (1.93–5.08)] (Figure 11).

Alleviation of Clinical symptoms.

Two studies, involving a total of 256 participants (129 in the astragalus group and 127 in the control group), were analyzed using a fixed-effects model due to the lack of significant heterogeneity in gas passage (*I*^2^ = 0%, *p* = 0.79) and defecation passage (*I*^2^ = 0%, *p* = 0.79). The meta-analysis results indicate that astragalus can reduce the recovery time for gas passage [*MD* = −3.53, 95%*CI* (−6.17~−0.88)] and defecation [*MD* = −3.56, 95%*CI* (−6.27~−0.85)] (Figure 12).

For the disappearance time of abdominal pain and distension, three studies with 256 patients were assessed using a random-effects model due to high heterogeneity. The analysis exhibits that astragalus decreases the disappearance time of abdominal pain [*MD* = −34.30, 95%*CI* (−60.08, −8.53)] and distension [*MD* = −25.57, 95%*CI* (−37.51, −13.63)] (Figure 13).

Recovery time of bowel sounds was evaluated in five studies with 512 patients, also using a random-effects model due to significant heterogeneity (*I*^2^ = 97%, *p* < 0.00001). The meta-analysis demonstrates that astragalus reduces the recovery time of bowel sounds [*MD* = −21.37, 95%*CI* (−37.29, −5.44)] (Figure 14).

Sensitivity analysis

A sensitivity analysis was performed using Stata 18 to assess the reliability of the meta-analytic findings regarding the therapeutic efficacy of astragalus in AP. The sensitivity analysis results exhibit that all six studies are encompassed within the 95% CI, indicating consistency across the included studies, and the meta-analysis results are not sensitive to the exclusion of individual studies (Figure 15). Furthermore, the heterogeneity analysis of astragalus’s therapeutic efficacy in AP is low (*I*^2^ = 7%, *p* = 0.37). Overall, the meta-analysis results exhibit considerable robustness.

Assessment of publication bias

The publication bias in studies reporting the efficacy of astragalus in treating AP was assessed using a funnel plot. All six studies were symmetrically distributed around the middle line and fell within the 95%*CI* interval, suggesting minimal publication bias (Figure 16).

## 4. Discussions

AP remains a significant abdominal disease with substantial morbidity and mortality across all age groups, leading to a profound impact on quality of life and imposing substantial socio-economic burdens through persistent pain and prolonged hospital stays [5]. There is an urgent need for novel pharmaceutical interventions and treatment modalities. Concurrently, traditional Chinese medicine exhibits considerable therapeutic potential for this condition in clinical settings, notably exemplified by various formulations of astragalus such as astragalus injection and combined decoctions with other traditional Chinese medicines [14,15]. However, astragalus’s therapeutic application in AP is limited by unclear pharmacological mechanisms and the absence of established clinical guidelines, meta-analyses, or systematic reviews.

The core targets identified in the PPI network analysis comprised TP53, AKT1, TNF, IL6, EGFR, CASP3, MYC, and HIF1A based on our findings. The progression of AP was exacerbated by the induction of pancreatic acinar cell apoptosis through the activation of the TP53 protein encoded by TP53 [16]. The in vitro PI3K-Akt pathway is crucial for trypsinogen activation, and inhibiting AKT1 expression can mitigate inflammatory damage caused by trypsinogen activation [17]. A decreased AKT1 expression was found to ameliorate lung inflammatory damage in AP, as demonstrated in an animal study [18]. TNF-α, a key immune marker of inflammation, was demonstrated to alleviate AP symptoms in both animal experiments and clinical trials upon inhibition of its activity [19,20]. IL-6 is released from necrotic pancreatic cells, contributing to systemic inflammatory responses and tissue injury by mediating neutrophil migration and activation of the JAK/STAT signaling pathway [21]. The downregulation of IL-6 expression levels using astragalus polysaccharides has been demonstrated to mitigate the inflammatory response in rat models of AP [22]. Hyperactivation of EGFR signaling leads to severe pancreatitis, whereas EGFR inhibition improves symptoms and reduces organ damage in AP rats [23,24]. The BAX/BCL2/CASP3 apoptosis signaling pathway is implicated in both the inflammatory response and apoptosis in AP [25]. Suppression of CASP3 expression via miR-339-3 has been demonstrated to alleviate AP [26]. Hyperactive MYC promotes inflammation and cancer transformation by inhibiting the negative regulation of Splicing Factor SRSF1 in AP [27]. HIF-1α triggers intra-pancreatic coagulation and increases the expression of amylase and lipase genes [28]. The reduction in serum HIF-1α levels with Huangqin decoction has been demonstrated to inhibit the inflammatory response in a clinical trial [29]. Given the significant roles of these key targets in the pathogenesis and treatment of AP, as revealed in previous studies, the modulation of these targets constitutes a pivotal aspect of astragalus’s pharmacological mechanism in treating AP.

To elucidate the pharmacological mechanism of astragalus in AP, we employed Network Pharmacology to identify the key bioactive compounds involved in this therapeutic process. Quercetin, kaempferol, isorhamnetin, formononetin, and calycosin were identified as the top five active compounds demonstrating potential interactions with core targets, as assessed by the molecular docking analysis. The pharmacological actions of quercetin have been documented to include the reduction in intracellular reactive oxygen species (ROS) production; endoplasmic reticulum (ER) stress response; and modulation of inflammatory factors, such as TNF-α, IL-6, and IL-10, as well as inhibition of the p38/MAPK signaling pathway by upregulating miR-216b [30,31]. Calycosin demonstrated efficacy by lowering TNF-α, IL-6, and IL-1β levels, suppressing myeloperoxidase (MPO) activity, enhancing superoxide dismutase (SOD) activity, and inhibiting NF-κB/p65 expression and the phosphorylation of IκBα and p38 MAPK, contributing to its therapeutic efficacy in AP [32]. Additionally, kaempferol and isorhamnetin mitigate mitochondrial damage and ROS accumulation, thereby alleviating pancreatic acinar cell necrosis in AP [33,34]. While previous research has elucidated some mechanisms of quercetin, kaempferol, isorhamnetin, and calycosin in the context of AP, our study identifies previously unreported potential mechanisms involving TP53, AKT1, EGFR, CASP3, MYC, and HIF1A and their associated signaling pathways. Notably, the effects and mechanisms of formononetin in AP remain unexplored. Further exploratory experiments are necessary to validate our findings regarding the components of astragalus in AP.

During astragalus treatment for AP, key biological processes were identified encompassing positive regulation of transcription by RNA polymerase II, negative regulation of apoptotic processes, and positive regulation of gene expression. Predominantly, cellular components are predominantly localized in the cytoplasm, nucleoplasm, and cytosol, while molecular functions are chiefly associated with protein binding, identical protein binding, and enzyme binding. Furthermore, the principal signaling pathways identified in this process include pathways in cancer, PI3K-Akt signaling pathway, and lipid metabolism and atherosclerosis. The PI3K-Akt signaling pathway is of particular interest. While numerous studies have documented its impact on AP [35,36], the effects of astragalus on AP through this pathway remain underreported. Based on the findings of this study, with a focus on the pathways in cancer, PI3K-Akt signaling pathway, lipid metabolism, and atherosclerosis pathways, further pharmacological mechanisms of astragalus in AP will be illuminated.

The meta-analysis aimed to validate the therapeutic efficacy of astragalus for AP. Our findings indicate that astragalus can effectively treat AP and alleviate clinical symptoms by reducing the interval time of gas or defecation passage, the disappearance time of abdominal pain or distension, and the recovery time of bowel sounds. However, this study has some limitations, the absence of double blinding in the included RCTs introduces a high risk of performance and detection bias, as reflected by Jadad scores indicating low to moderate quality. Consequently, further high-quality, double-blind randomized controlled trials are necessary to validate the therapeutic efficacy of astragalus for AP. Despite these limitations, efforts to ensure result accuracy involved employing a unique analysis model to address study heterogeneity. Moreover, the funnel plot analysis indicated minimal publication bias, and sensitivity analysis demonstrated that all six studies fell within the 95% confidence interval, underscoring the robustness of the meta-analysis results.

This study serves as preliminary work using network pharmacology and molecular docking approaches. Further in vitro and in vivo experimental validation will be conducted to support the proposed mechanisms. We plan to construct an AP cell model and animal model induced by Coxsackievirus B group 3 (CVB3) using rhesus monkeys’ pancreatic acinar cells and rhesus monkeys, respectively. Beyond gallstones, hyperlipidemia, and alcoholism, AP can also be triggered by viruses including Coxsackievirus, hepatitis virus, mumps virus, human immunodeficiency virus, and Echovirus [37]. Of these, CVB3 and CVB4 are most frequently detected in Coxsackievirus [38]. The therapeutic efficacy and pharmacological mechanisms of astragalus injection in virus-induced AP remain unclear. Thus, we aim to explore the efficacy and mechanisms of astragalus injection in CVB3-induced AP. Serum amylase (AMS) and lipase (LPS) levels, critical clinical markers of AP, alongside pancreatic acinar cell lesions or inflammatory changes, will be used to confirm the successful establishment of the cell and animal models. Following astragalus injection, its therapeutic efficacy in CVB3-induced AP will be evaluated by analyzing changes in AMS, LPS, and inflammatory pathology using quantitative PCR, Western blot, and histological analyses of the rhesus monkey pancreas. The changes in the core target proteins and their signaling pathways will be investigated to elucidate the pharmacological mechanisms of astragalus injection in this condition.

In summary, the pharmacological mechanism of astragalus in treating AP entails the synergistic action of multiple components, targets, and pathways. Key active ingredients, such as quercetin, kaempferol, isorhamnetin, formononetin, and calycosin, interact with core targets, including TP53, AKT1, TNF, IL6, EGFR, CASP3, MYC, and HIF1A, within primary pathways like Pathways in cancer, Lipid metabolism and atherosclerosis, and the PI3K-Akt signaling pathway. Particularly, Quercetin exhibits potential synergy with these core targets. Astragalus effectively treats AP and alleviates clinical symptoms. This study elucidates the potential pharmacological targets and underlying mechanisms of astragalus in AP and further confirms its therapeutic efficacy through a meta-analysis, offering some insights for its therapeutic application.

## Figures and Tables

**Figure 1 cimb-47-00379-f001:**
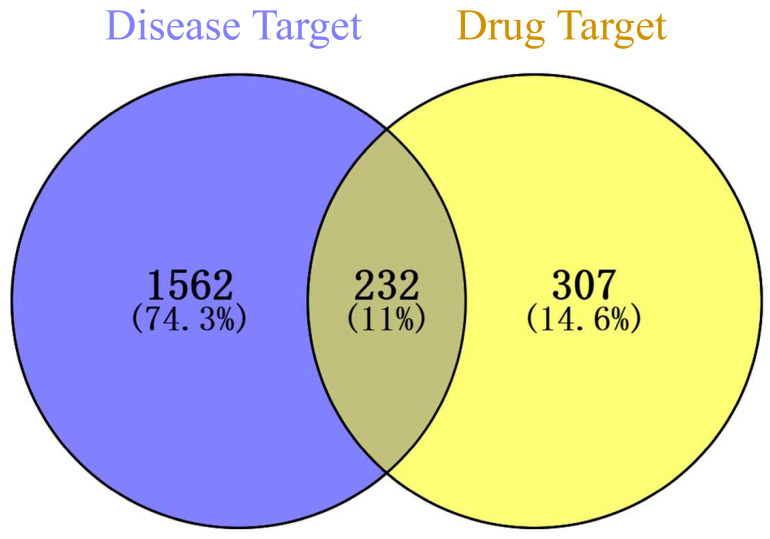
Therapeutic targets of astragalus in AP treatment.

**Figure 2 cimb-47-00379-f002:**
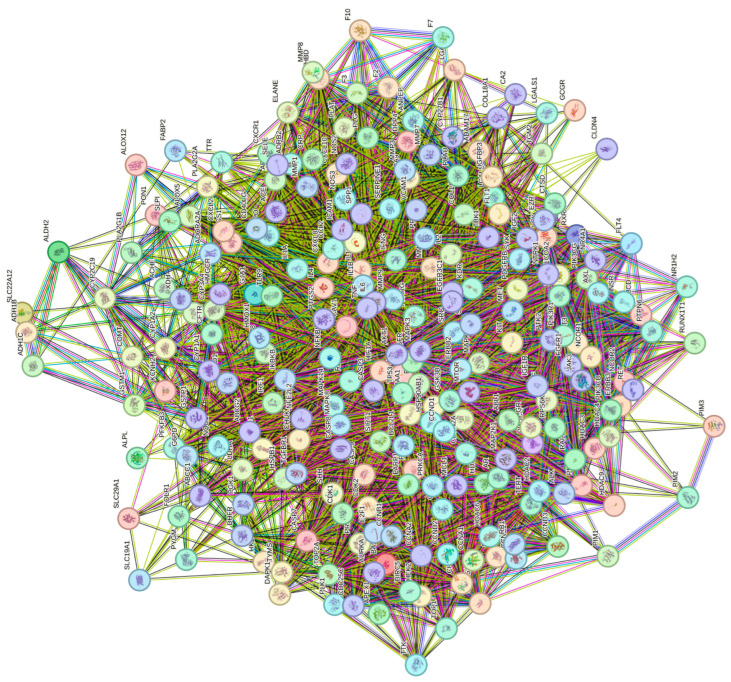
Protein–protein interaction network.

**Figure 3 cimb-47-00379-f003:**
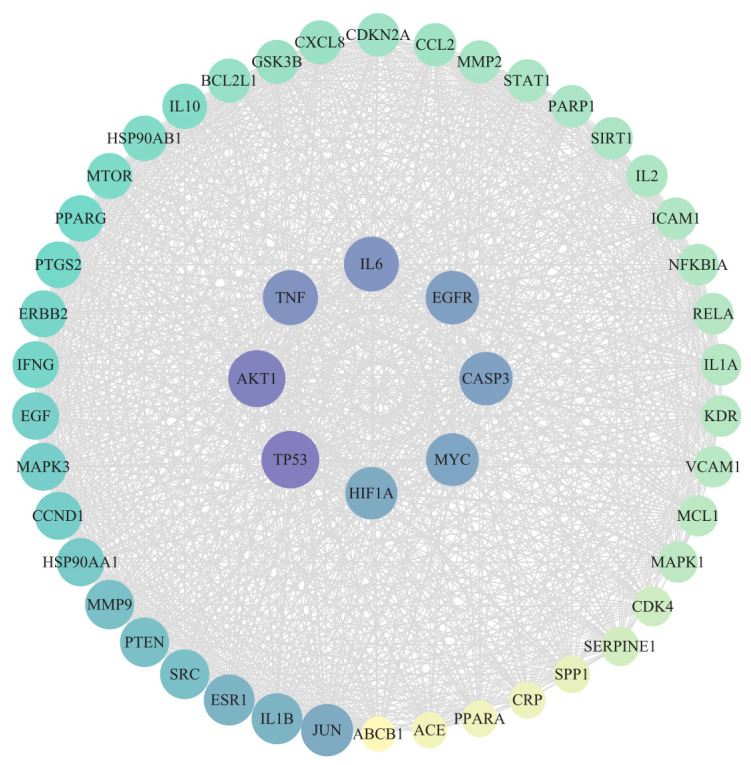
Core targets interaction network.

**Figure 4 cimb-47-00379-f004:**
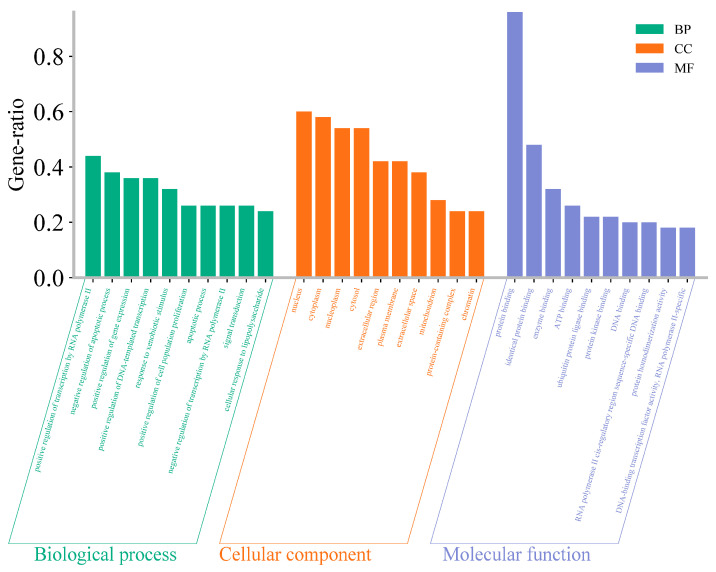
Three-in-one histogram of GO enrichment analysis.

**Figure 5 cimb-47-00379-f005:**
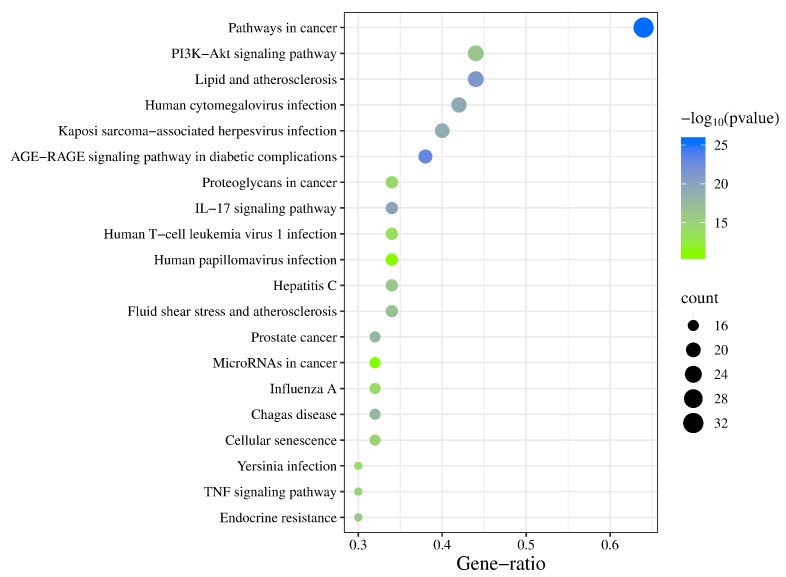
Bubble diagram of the KEGG pathway analysis.

**Figure 6 cimb-47-00379-f006:**
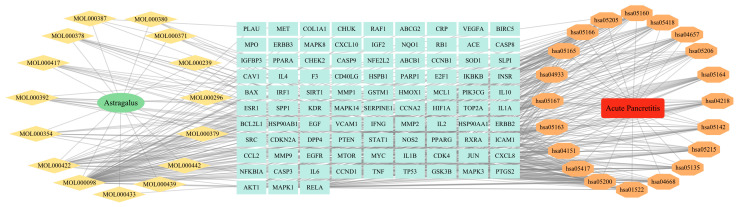
Drug–ingredients–targets–pathways–disease topological network.

**Figure 7 cimb-47-00379-f007:**
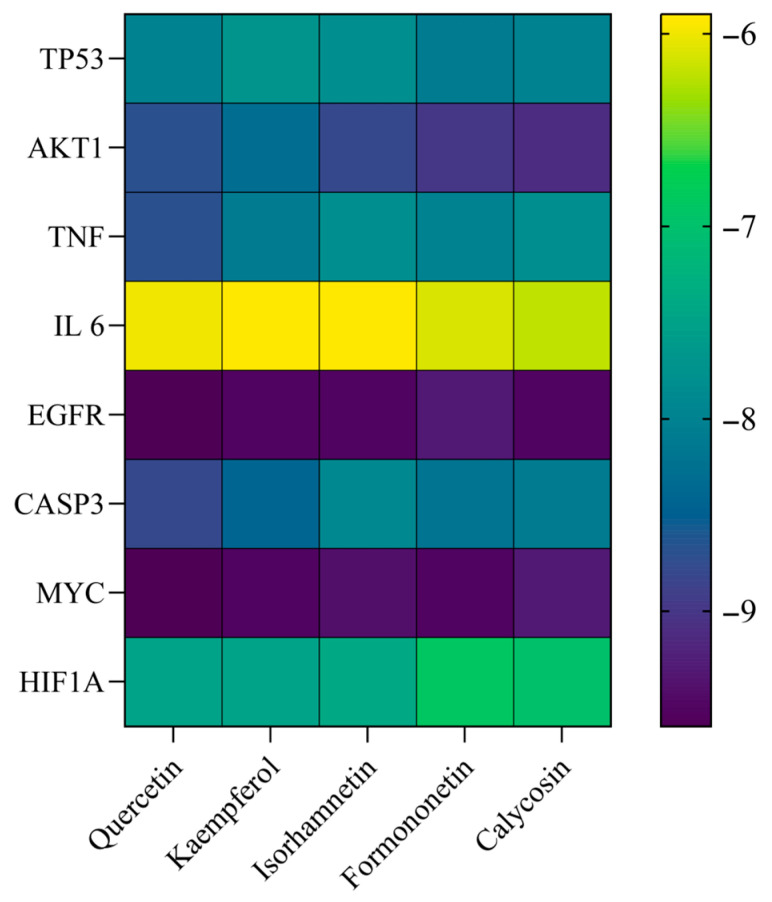
Heat map of the molecular docking results.

**Figure 8 cimb-47-00379-f008:**
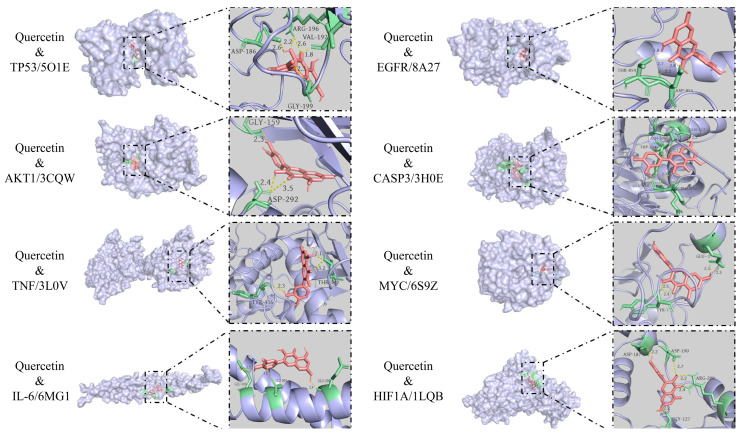
Pattern diagrams of the molecular docking results.

**Figure 9 cimb-47-00379-f009:**
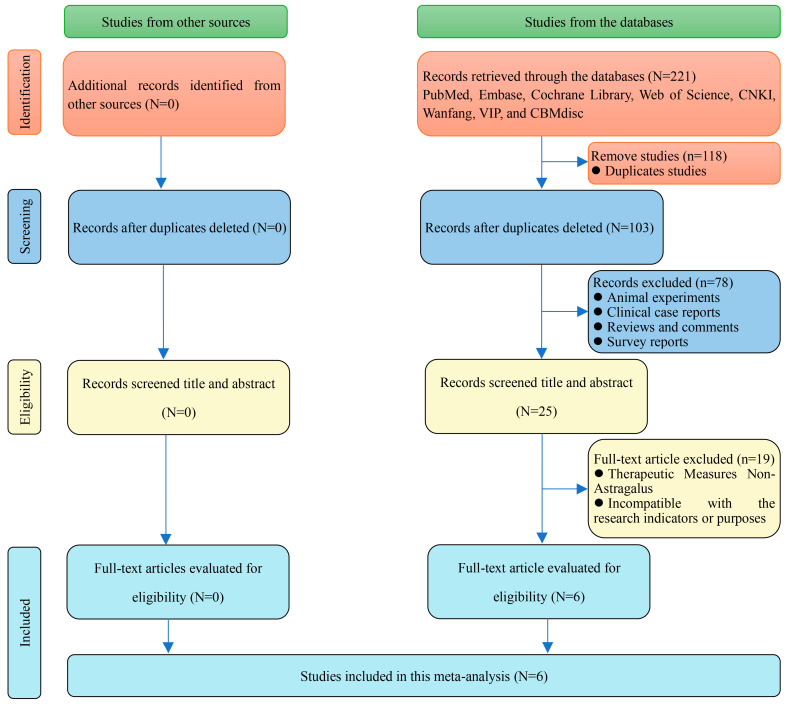
Literature search flow chart.

**Figure 10 cimb-47-00379-f010:**
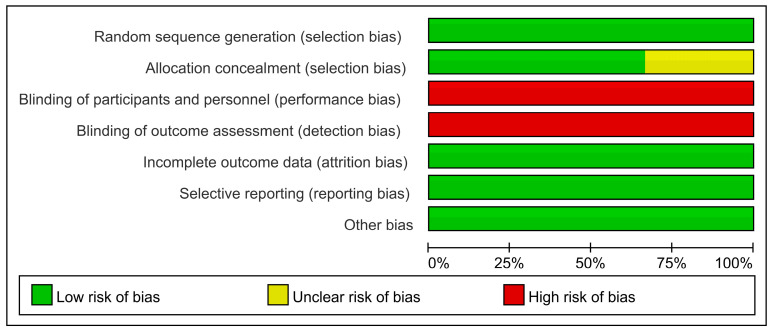
Risk of bias graph.

**Figure 11 cimb-47-00379-f011:**
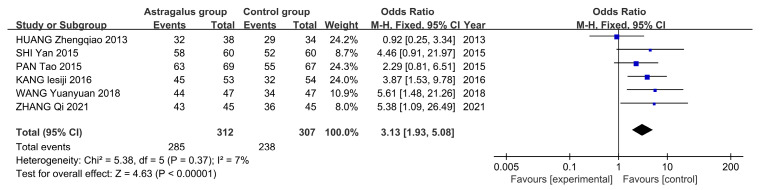
Forest plot of astragalus’s treatment efficacy [9,10,11,12,13,14].

**Figure 12 cimb-47-00379-f012:**
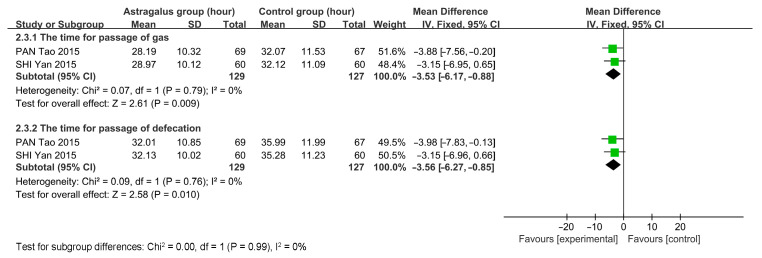
Forest plots for the time of gas and defecation passage [12,13].

**Figure 13 cimb-47-00379-f013:**
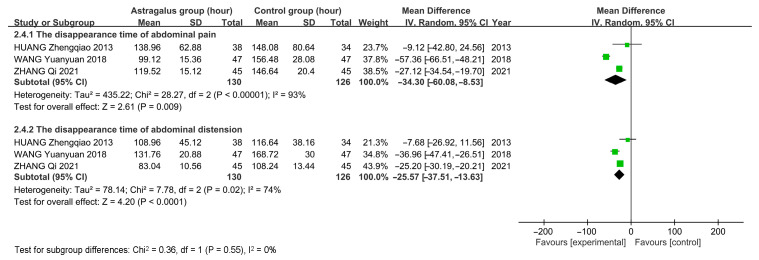
Forest plots for disappearance time of abdominal pain and distension [10,14,15].

**Figure 14 cimb-47-00379-f014:**
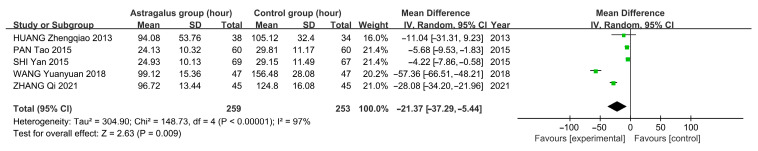
Forest plots for recovery time of bowel sounds [10,12,13,14,15].

**Figure 15 cimb-47-00379-f015:**
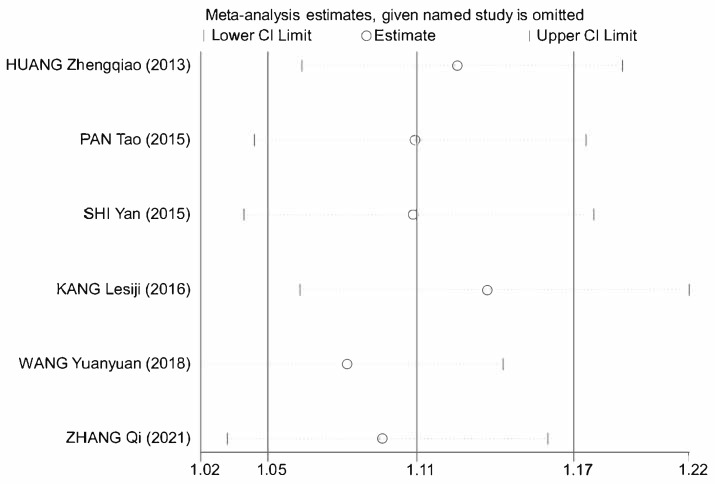
Sensitivity analysis plot of astragalus’s treatment efficacy [10,11,12,13,14,15].

**Figure 16 cimb-47-00379-f016:**
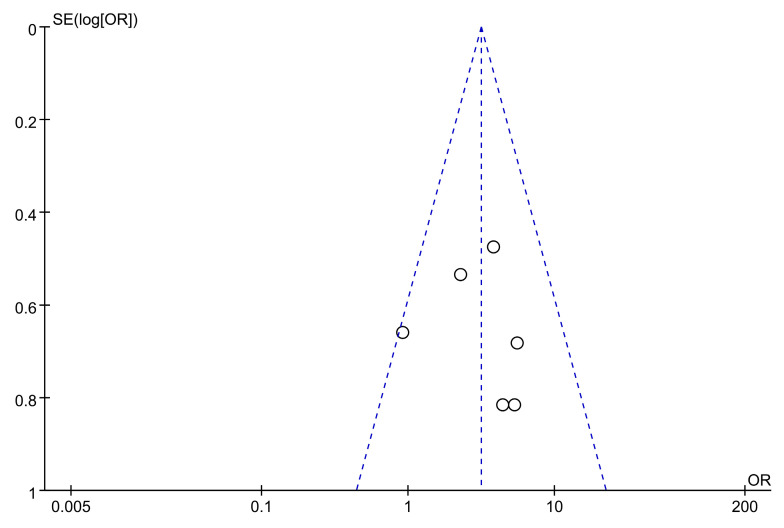
Funnel plot of the publication bias.

**Table 1 cimb-47-00379-t001:** Active ingredients of astragalus.

Code	Ingredient	OB (Oral Bioavailability) (%)	DL (Drug-Likeness)
MOL000392	Formononetin	69.67	0.21
MOL000422	Kaempferol	41.88	0.24
MOL000417	Calycosin	47.75	0.24
MOL000438	(3R)-3-(2-hydroxy-3,4-dimethoxyphenyl)chroman-7-ol	67.67	0.26
MOL000098	Quercetin	46.43	0.28
MOL000239	Jaranol	50.83	0.29
MOL000398	Isoflavanone	109.99	0.3
MOL000378	7-O-methylisomucronulatol	74.69	0.3
MOL000354	Isorhamnetin	49.6	0.31
MOL000380	(6aR,11aR)-9,10-dimethoxy-6a,11a-dihydro-6H-benzofurano[3,2-c]chromen-3-ol	64.26	0.42
MOL000371	3,9-Di-O-methylnissolin	53.74	0.48
MOL000442	1,7-Dihydroxy-3,9-dimethoxy pterocarpene	39.05	0.48
MOL000439	Isomucronulatol-7,2′-di-O-glucosiole	49.28	0.62
MOL000387	Bifendate	31.1	0.67
MOL000374	5′-Hydroxyiso-muronulatol-2′,5′-di-O-glucoside	41.72	0.69
MOL000433	FA	68.96	0.71
MOL000296	Hederagenin	36.91	0.75
MOL000211	Mairin	55.38	0.78
MOL000033	(3S,8S,9S,10R,13R,14S,17R)-10,13-Dimethyl-17-[(2R,5S)-5-propan-2-yloctan-2-yl]-2,3,4,7,8,9,11,12,14,15,16,17-Dodecahydro-1H-cyclopenta[a]phenanthren-3-ol	36.23	0.78
MOL000379	9,10-Dimethoxypterocarpan-3-O-β-D-glucoside	36.74	0.92

**Table 2 cimb-47-00379-t002:** The core target of astragalus in the treatment of AP.

Gene Name	Degree	Betweenness	Closeness
TP53	185	2422.629693	0.003610108
AKT1	182	1795.229331	0.003571429
TNF	171	1963.275514	0.003436426
IL6	170	1653.318961	0.003424658
EGFR	163	1693.215029	0.003344482
CASP3	162	994.273886	0.003322259
MYC	159	1536.505978	0.00330033
HIF1A	156	1012.411402	0.003267974

**Table 3 cimb-47-00379-t003:** Molecular docking results of quercetin and key targets.

Ligand Compound	Receptor Protein	PUB (ID)	Binding Energy (KJ/mol)	Positive Control Ligand	Binding Energy (KJ/mol)
Quercetin	TP53	5O1E	−8.0	5o1e_D_9GT	−7.1
Quercetin	AKT1	3CQW	−8.6	3cqw_D_CQW	−8.9
Quercetin	TNF	3L0V	−8.7	3l0v_D_724	−8.6
Quercetin	IL-6	6MG1	−6.0	6mg1_I_GOL	−2.9
Quercetin	EGFR	8A27	−9.6	8a27_C_KY9	−10.4
Quercetin	CASP3	3H0E	−8.8	3h0e_C_H0E	−8.6
Quercetin	MYC	6S9Z	−9.6	6s9z_H_VKL	−8.7
Quercetin	HIF1A	1LQB	−7.5	1lqb_E_SO4	−3.4

**Table 4 cimb-47-00379-t004:** Basic characteristic of included studies.

First Author and Publication Year	Number of Participants Astragalus and Control	Intervention		Frequency and Treatment Duration (d)	Jadad Score	Research Indicators
		Astragalus Group	Control Group			
HUANG Zhengqiao 2013 [10]	38 & 34	General Treatment and Astragalus	General Treatment and Metoclopramide Injection Subcutaneous 2 mL	1–2 treatments per day for 2 days	2	①④⑤⑥
PAN Tao 2015 [11]	69 & 67	General treatment and astragalus injection subcutaneous 2 mL	General treatment and saline (medicine) subcutaneous 2 mL	2 treatments per day for 7 days	3	①②③⑥
SHI Yan 2015 [12]	60 & 60	General treatment and astragalus injection subcutaneous 0.5~1 mL	General treatment and saline (medicine) subcutaneous 0.5~1 mL	1–2 times daily for 20 treatments	3	①②③⑥
KANG Lesiji 2016 [13]	53 & 54	General treatment and astragalus injection IV 20 mL	General treatment	1 treatment per day for 7 days	2	①
WANG Yuanyuan 2018 [14]	47 & 47	General treatment and astragalus injection IV 60 mL	General treatment	1 treatment per day for 10 days	4	①④⑤⑥
ZHANG Qi 2021 [15]	45 & 45	General treatment and astragalus injection IV 20 mL	General treatment	1 treatment per day for 10 days	3	①④⑤⑥

Note: ① Treatment effect; ② time to gas passage; ③ time to defecation passage; ④ disappearance time of abdominal pain; ⑤ disappearance time of abdominal distension; and ⑥ recovery time of bowel sounds.

## Data Availability

Data are contained within the article and Appendix A.

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
