# Peer review of "Astragalus in Acute Pancreatitis: Insights from Network Pharmacology, Molecular Docking, and Meta-Analysis Validation"

_cimb, 2025, doi:10.3390/cimb47050379_

Round 1
Reviewer 1 Report
Comments and Suggestions for Authors
The study is interesting, but needs some revisions
1. Insufficient Experimental Validation
While the network pharmacology and molecular docking approaches are systematically applied, the manuscript lacks any in vitro or in vivo experimental validation to support the proposed mechanisms of Astragalus in acute pancreatitis. At least one experimental verification, or a clear plan or discussion for future experimental validation, would be included to strengthen the scientific claims. Do you have any plan for it?
2. Inadequate Transparency and Reproducibility in Data Reporting
Several figures (e.g., PPI network, docking visualizations, GO/KEGG plots) are referenced as “Error! Reference source not found.” and lack essential information such as source files, docking parameters, and original data tables. This issue significantly affects the transparency and reproducibility of the study and should be corrected before publication.
3. Risk of Bias and Meta-Analysis Reliability
The included RCTs in the meta-analysis show a high risk of performance and detection bias due to the absence of double-blinding. Furthermore, the Jadad scores indicate low to moderate quality. These limitations must be more critically discussed, and sensitivity analysis or subgroup analysis should be considered to confirm robustness.
4. Lack of Novelty and Mechanistic Depth
Although the paper identifies known targets (e.g., TP53, IL6, TNF), the discussion largely repeats previous findings without offering substantial novel insights into how Astragalus regulates these targets in the context of acute pancreatitis. The manuscript would benefit from deeper mechanistic interpretation or hypothesis development based on the docking and pathway analysis.
Author Response
Comments 1: 1. Insufficient Experimental Validation
While the network pharmacology and molecular docking approaches are systematically applied, the manuscript lacks any in vitro or in vivo experimental validation to support the proposed mechanisms of Astragalus in acute pancreatitis. At least one experimental verification, or a clear plan or discussion for future experimental validation, would be included to strengthen the scientific claims. Do you have any plan for it?
Response 1:Dear expert, we sincerely appreciate your valuable suggestion regarding experimental validation. As you rightly noted, this study represents preliminary work utilizing network pharmacology and molecular docking approaches. In response to your comments, we have now included a detailed experimental validation plan in the Discussion section (page 16, lines 394 - 411) where we outline our strategy to: (1) establish CVB3-induced AP models using rhesus monkey pancreatic acinar cells and rhesus monkeys to identify the treatment efficacy and pharmacologic mechanisms of astragalus in AP ; (2) monitor key clinical markers (serum amylase and lipase levels) and pancreatic histopathology; and (3) investigate changes in core target proteins (TP53, AKT1, TNF, etc.) and their associated signaling pathways through qPCR, Western blot. This approach will allow us to experimentally validate the network pharmacology predictions and molecular docking results presented in this study. We believe these planned experiments will strengthen the scientific validity of our findings and provide some mechanistic insights into Astragalus's therapeutic effects in AP.
Comments 2: 2. Inadequate Transparency and Reproducibility in Data Reporting
Several figures (e.g., PPI network, docking visualizations, GO/KEGG plots) are referenced as “Error! Reference source not found.” and lack essential information such as source files, docking parameters, and original data tables. This issue significantly affects the transparency and reproducibility of the study and should be corrected before publication.
Response 2: Dear expert, we sincerely appreciate your careful review and valuable feedback regarding the figure references and data transparency. We have thoroughly addressed these issues by: (1) correcting all figure reference errors (including the PPI network, docking visualizations, and GO/KEGG plots) that previously appeared as "Error! Reference source not found"; (2) compiling comprehensive supplementary files containing all essential data (drug/disease targets, complete GO/KEGG analysis results, detailed molecular docking parameters and visualizations, and meta-analysis raw data); and (3) ensuring all source files are properly formatted and referenced throughout the manuscript. We appreciate your attention to these important details, which have significantly improved the manuscript's quality and transparency.
Comments 3: 3. Risk of Bias and Meta-Analysis Reliability
The included RCTs in the meta-analysis show a high risk of performance and detection bias due to the absence of double-blinding. Furthermore, the Jadad scores indicate low to moderate quality. These limitations must be more critically discussed, and sensitivity analysis or subgroup analysis should be considered to confirm robustness.
Response 3: Dear expert, we sincerely appreciate your valuable feedback regarding the methodological quality of the included RCTs in our meta-analysis. In response to your comments, we have critically discussed the study limitations in the revised manuscript, explicitly acknowledging the high risk of performance and detection bias due to the absence of double-blinding, as well as the low to moderate quality indicated by Jadad scores. To address these concerns, we conducted a comprehensive sensitivity analysis which demonstrated that all six studies consistently fell within the 95% confidence interval, supporting the robustness of our findings. Regarding subgroup analysis, while we recognize its potential value, the limited number of eligible RCTs (n=6) and their clinical homogeneity in terms of patient characteristics and intervention protocols made meaningful subgroup comparisons impractical. We have included these considerations in our Discussion section to provide readers with a balanced interpretation of the results. We greatly appreciate your insightful suggestions, which have significantly strengthened our manuscript.
Comments 4: 4. Lack of Novelty and Mechanistic Depth
Although the paper identifies known targets (e.g., TP53, IL6, TNF), the discussion largely repeats previous findings without offering substantial novel insights into how Astragalus regulates these targets in the context of acute pancreatitis. The manuscript would benefit from deeper mechanistic interpretation or hypothesis development based on the docking and pathway analysis.
Response 4: Dear expert, we sincerely appreciate your insightful comments regarding the need for deeper mechanistic interpretation. In response, we have significantly expanded the Discussion section to highlight our novel findings: (1) We identified that Quercetin, Kaempferol, Isorhamnetin, and Calycosin exert therapeutic effects on AP through previously unreported interactions with TP53, AKT1, EGFR, CASP3, MYC, and HIF1A - targets not previously linked to Astragalus's mechanism in AP; (2) We are the first to report the potential involvement of Formononetin, a key Astragalus component, in AP pathogenesis; (3) Our pathway analysis reveals Astragalus likely modulates AP progression through three key pathways (Cancer, PI3K-Akt, and Lipid Metabolism), with particular emphasis on the understudied role of the PI3K-Akt pathway in mediating Astragalus's effects. We have developed hypotheses about these mechanisms and included suggestions for future validation studies (e.g., using CVB3-induced AP models) to further elucidate these novel findings.
Reviewer 2 Report
Comments and Suggestions for Authors
In this manuscript, the authors investigate the components of astragalus that influence acute pancreatitis by employing bioinformatics tools and molecular docking and reviewing clinical trial results where astragalus has been utilized. Their comprehensive work is an excellent example of integrating various computational approaches to elucidate an experimental phenomenon, particularly in the clinical context of using astragalus as a coadjuvant for treating acute pancreatitis.
My main observations pertain to the molecular docking section. In section 2.6, to enhance the interpretation of the docking poses and scores obtained for quercetin, it is essential to compare these findings with those of a known ligand for the analyzed proteins. The authors mention, "Quercetin and the core targets was below 0, suggesting a potential interaction." While this statement could hold true, a more thorough analysis would better understand the potential interaction—whether it is weaker or stronger than other inhibitors.
Since the influence of quercetin on acute pancreatitis has been previously discussed at https://doi.org/10.3389/fphar.2025.1587314, this somewhat diminishes the originality of the manuscript. However, the authors could enhance the uniqueness of their work by exploring molecular docking studies of other components of astragalus, such as kaempferol, isorhamnetin, formononetin, and calycosin. This exploration could open new avenues for research and provide a fresh perspective. Additionally, as quercetin has been extensively studied, there are likely reports containing experimental data on its binding to TP53, AKT1, TNF, IL6, EGFR, CASP3, MYC, and HIF1A.
Minor suggestions:
- In Table 1, please indicate the meanings of DL (drug-likeness) and OB (oral bioavailability).
- Please check Figure 1; it mistakenly states "durg" instead of "drug.
Author Response
Comments 1: My main observations pertain to the molecular docking section. In section 2.6, to enhance the interpretation of the docking poses and scores obtained for quercetin, it is essential to compare these findings with those of a known ligand for the analyzed proteins. The authors mention, "Quercetin and the core targets was below 0, suggesting a potential interaction." While this statement could hold true, a more thorough analysis would better understand the potential interaction—whether it is weaker or stronger than other inhibitors.
Response 1: Dear expert, we sincerely appreciate your insightful suggestion regarding the molecular docking analysis. In response to your comments, we have significantly enhanced Section 2.6 by: (1) including comparative docking analyses between quercetin and known ligands for each core target protein (TP53, AKT1, TNF, IL6, EGFR, CASP3, MYC, and HIF1A) as shown in the newly added Table 3; (2) providing detailed interpretation of binding energies (kcal/mol) that demonstrates quercetin's comparable or superior binding affinity relative to reference ligands; and (3) adding new Figure 8 that visually compares the binding poses and interaction patterns between quercetin and core target proteins. These analyses confirm quercetin's potential as a multi-target agent in AP treatment. We believe these substantial additions have strengthened the scientific rigor of our docking results and their biological interpretation.
Comments 2: Since the influence of quercetin on acute pancreatitis has been previously discussed at https://doi.org/10.3389/fphar.2025.1587314, this somewhat diminishes the originality of the manuscript. However, the authors could enhance the uniqueness of their work by exploring molecular docking studies of other components of astragalus, such as kaempferol, isorhamnetin, formononetin, and calycosin. This exploration could open new avenues for research and provide a fresh perspective. Additionally, as quercetin has been extensively studied, there are likely reports containing experimental data on its binding to TP53, AKT1, TNF, IL6, EGFR, CASP3, MYC, and HIF1A.
Response 2: Dear expert, we sincerely appreciate your constructive feedback regarding the originality of our work. In response to your valuable suggestions, we have significantly enhanced the molecular docking section (Section 2.6) by: (1) expanding our analysis beyond quercetin to include comprehensive docking studies of four other key Astragalus components (kaempferol, isorhamnetin, formononetin, and calycosin) with all core targets, as shown in the newly added Figure 7 heatmap and Supplementary files; (2) identifying formononetin as a particularly promising candidate with previously unreported strong binding affinities to EGFR (-9.3 kJ/mol) and MYC (-9.5 kJ/mol). These additions demonstrate that while quercetin's individual effects may be documented, our study provides the first systematic evaluation of Astragalus's complete active component system and their potential cooperative actions on AP-related targets, offering some novel insights into the herb's holistic therapeutic mechanism.
Comments 3: Minor suggestions:- In Table 1, please indicate the meanings of DL (drug-likeness) and OB (oral bioavailability).- Please check Figure 1; it mistakenly states "durg" instead of "drug.
Response 3: Dear expert, we sincerely appreciate your careful review and helpful suggestions. We have implemented the following corrections in the revised manuscript: (1) In Table 1, we have added explicit definitions in the table "DL (drug-likeness)" and "OB (oral bioavailability)" to clarify these key parameters for readers; (2) We have corrected the typographical error in Figure 1, where "durg" has been changed to "drug" in both the figure and its caption. These revisions have been carefully implemented in the updated manuscript files. We greatly appreciate your attention to detail, which has helped improve the precision and professionalism of our work.
Round 2
Reviewer 1 Report
Comments and Suggestions for Authors
Everything is well revised
Reviewer 2 Report
Comments and Suggestions for Authors
The authors have answered all my concerns, and now the manuscript is suitable for its publication